# The Development of New Methods to Stimulate the Production of Antimicrobial Peptides in the Larvae of the Black Soldier Fly *Hermetia illucens*

**DOI:** 10.3390/ijms242115765

**Published:** 2023-10-30

**Authors:** Atsuyoshi Nakagawa, Takuma Sakamoto, Michael R. Kanost, Hiroko Tabunoki

**Affiliations:** 1Future Tech Laboratory, Corporate Research & Development, UBE Corporation, 8-1 Goi-Minamikaigan, Chiba 290-0045, Japan; a.nakagawa@ube.com; 2Cooperative Major in Advanced Health Science, Graduate School of Bio-Applications and System Engineering, Tokyo University of Agriculture and Technology, Tokyo 183-8509, Japan; 3Department of Science of Biological Production, Graduate School of Agriculture, Tokyo University of Agriculture and Technology, 3-5-8 Saiwai-Cho, Tokyo 183-8509, Japan; tsakamoto@go.tuat.ac.jp; 4Department of Biochemistry and Molecular Biophysics, Kansas State University, 141 Chalmers Hall, Manhattan, KS 66506-3702, USA; kanost@ksu.edu

**Keywords:** black soldier fly, antimicrobial peptides (AMPs), thermal injury, Gram-positive bacteria

## Abstract

(1) The global population is projected to reach a staggering 9.8 billion people by the year 2050, leading to major concerns about food security. The necessity to increase livestock production is inevitable. The black soldier fly (BSF) is known for its ability to consume a wide range of organic waste, and BSF larvae have already been used as a partial substitute for fishmeal. In contrast, the use of antibiotics in livestock feed for growth promotion and prophylaxis poses a severe threat to global health owing to antimicrobial resistance. Insect antimicrobial peptides (AMPs) have shown the potential to rapidly disrupt target bacterial membranes, making bacterial resistance to AMPs a less likely concern. (2) In this study, we explored various methods for stimulating AMP synthesis in BSF larvae and found that thermal injury effectively induced the production of various AMP types. Additionally, we investigated the activation of innate immune response pathways that lead to AMP production following thermal injury. (3) Interestingly, thermal injury treatment, although not involving bacteria, exhibited a similar response to that observed following Gram-positive bacterial infection in eliciting the expression of AMP genes. (4) Our findings offer support for the industrial use of BSF to enhance livestock production and promote environmental health.

## 1. Introduction

The black soldier fly (BSF; *Hermetia illucens*) is a dipteran insect known for its distinctive characteristics. BSF larvae can consume a wide range of organic waste materials, including food scraps from various sources and animal excrement [1]. Owing to these unique attributes, the BSF has garnered attention as a promising organism for the sustainable production of protein resources from organic waste [2]. With the global population projected by the United Nations to reach 9.77 billion by 2050 [3], the challenge of ensuring adequate food supply for this growing population is becoming increasingly important. To address this issue, there is a growing need to enhance livestock production, which, in turn, raises concerns about potential shortages in livestock feed.

In Japan, various livestock, such as cows, pigs, and chickens, are traditionally reared on a diet composed of a mixture of concentrate (comprising soybean, corn, wheat, fishmeal, skim milk, and other components) and roughage (including fresh grass, silage, hay, and straw) [4]. In recent years, BSF larvae have been used as a partial substitute for fishmeal in livestock feed [5,6,7]. Furthermore, the routine use of antibiotics in livestock farming, aimed at promoting growth, has become a topic of increasing concern [8,9,10,11]. Excessive antibiotic use poses serious societal risks, as it contributes to the emergence of antibiotic-resistant bacteria and their subsequent spread in the environment, jeopardizing public health [12]. Consequently, efforts are underway to identify alternative solutions that can replicate the growth-promoting effects of antibiotics while mitigating these associated risks.

One aspect of the insect humoral immune system is the fat body and hemocytes synthesis of antimicrobial peptides (AMPs) and lectins [13], which are then released into the hemolymph [14]. Insect AMPs are crucial antimicrobial defense molecules with distinct structural characteristics that enable the permeation and disruption of target membranes [14,15]. AMPs have an adverse effect on bacterial pathogens by disrupting their microbial membrane, a process that may be a barrier to the evolution of resistance to peptides [14]. Bacterial species utilized for immune challenge in BSF induce the synthesis of various sets of AMPs [16]. Therefore, BSF might serve as a rich reserve of AMPs with potent activities [17,18,19]. The humoral immune system of insects can be activated by the injection of living or dead bacteria [17,19,20]. However, injection of bacteria would not be a suitable method to produce larvae with high AMP expression for the purpose of feeding insect meal to livestock [21,22]. Another method to induce AMP production in larvae is by stab wounding the body. However, the AMPs generated after this treatment may become degraded during feed processing or stock preparation due to larval body damage. Therefore, there is a need for a safer and easier method to induce AMPs in BSF larvae.

Porcine beta-defensin-2 (pBD2) is an AMP belonging to the defensin family that is secreted from the intestinal tract in pigs. Oral administration of pBD2 to piglets can improve the feed conversion ratio [23,24]. Another AMP, cecropin A(1-11)-D(12-37)-Asn (CADN), based on the cecropin family from insects, can enhance broilers’ feed conversion ratio upon oral administration [25]. Considering these findings, insect AMPs may function as a potent substitute for antibiotics [15,16,17,18]. AMPs disrupt the bacterial plasma membrane and then make pores, resulting in the leakage of intracellular contents through permeabilization. This mechanism occurs within minutes, and as a result, AMPs can kill pathogens rapidly [26,27]. These natural AMPs have an added advantage in that they have lower chances of selecting for the emergence of resistance in pathogens because they act on lipid membranes, a fundamental cell structure, rather than on proteins, which may be modified more easily by mutation.

In this study, we investigated various types of stimulation for AMP induction in BSF larvae, and we analyzed possible innate immune response pathways leading to the expression of AMPs following thermal injury. 

## 2. Results

### 2.1. Thermal Injury Treatment Induces AMP Gene Expression

We concentrated on two representative AMPs in BSF, defensin-like peptide 4 (HiDLP4) and cecropin-like peptide 1 (HiCLP1). An increased production of HiDLP4 and HiCLP1 due to the immune response against *Staphylococcus aureus* has already been demonstrated in BSF [17,20].

First, we cloned the cDNAs of HiDLP4 and HiCLP1 for validation of their nucleotide sequence. HiDLP4 and HiCLP1 nucleotide sequences were submitted to the GenBank/DDBJ SAKURA Data bank under accession numbers LC741404 (for HiDLP4) and LC741405 (for HiCLP1). Then, the mRNA expression of HiDLP4 and HiCLP1 was evaluated using RT-qPCR analysis. We exposed the last instar larvae to various types of stimulation to identify alternative strategies without using bacteria for eliciting AMP production. We examined the production of AMPs following HCl treatment, heat treatment, stab wounding, thermal injury, and *Micrococcus luteus* injection in BSF larvae. The thermal injury, stab wound, and *M. luteus* injection groups exhibited significantly increased expression of HiDLP4 and HiCLP1 mRNA (Figure 1 and Figure 2a–c). In contrast, the HCl treatment and heat treatment groups did not exhibit increased HiDLP4 and HiCLP1 mRNA expression (Figure 1 and Figure 2d,e).

### 2.2. Thermal Injury Inhibits the Growth of Micrococcus luteus

Next, we examined the antimicrobial activity against the Gram-positive bacterium *M. luteus* in the hemolymph after thermal injury, stab wounding, or *M. luteus* injection. 

Park et al. reported that HiDLP4 exerts antimicrobial activity against the Gram-positive bacterium methicillin-resistant *S. aureus* [20]. Hence, we employed a Gram-positive bacterium, *M. luteus*, to characterize antimicrobial activity.

Typically, bacterial growth in culture has four phases: lag phase, log phase, stationary phase, and death phase. Once the bacterial growth phase finishes the lag phase and reaches the log phase, bacterial cell counts double continuously until nutrient depletion. We observed the time in the lag phase, with a delay in reaching the log phase, as an indication of antibacterial activity.

The time of the lag phase significantly extended in the ampicillin group, which is the positive control for the Gram-positive bacterium (Figure 3). Hemolymph collected at 9 (*p* = 0.0123), 15 (*p* = 0.0035), and 18 (*p* = 0.0018) h after thermal injury extended the time in the lag phase compared with the non-stimulus control group (Figure 3a,d). Hemolymph collected after a stab wound also extended the time of the lag phase. However, the antibacterial activity at 9 (*p* = 0.3105), 15 (*p* = 0.4512), 18 (*p* = 0.8512), 21 (*p* = 0.2564), and 24 (*p* = 0.2721) h after stab wounds did not differ from that observed in the thermal injury group (Figure 3a,b). In contrast, the time of the lag phase significantly increased at 24 h after *M. luteus* injection compared to the 24 h thermal injury group (Figure 3c, *p* = 0.00017).

### 2.3. Identification of Differentially Expressed Genes and Assignment of Drosophila melanogaster Homologs

We investigated possible mechanisms for AMP production after thermal injury. First, we examined HiDLP4 and HiCLP1 mRNA expression at 3, 6, 9, and 24 h. We found that the peak of mRNA expression of HiDLP4 and HiCLP1 occurred at 6 h (Figure 4a) and 3 h (Figure 4b) after thermal injury, respectively. Next, we assigned BSF transcripts to *Drosophila melanogaster* proteins to annotate potential gene function by blastx with a cut-off E-value of 1 × 10^–10^. Three non-stimulus control samples and thermal injury samples collected at 3, 6, and 9 h after thermal injury were subjected to RNA-seq analysis (Appendix A). The RNA-seq data were then mapped with HISAT2 and StringTie. From these RNA-seq data, we were able to extract 36,413 transcripts, out of which 28,583 (78.5%) of the BSF genes showed homology with *D. melanogaster* genes. Then, in comparison to the non-stimulus control groups, we extracted the genes that were differentially expressed in the 3, 6, and 9 h after thermal injury groups (Figure 5a–c). We discovered 420 differentially expressed genes in the thermal injury group after 3 h (Figure 5a), 388 differentially expressed genes after 6 h (Figure 5b), and 349 differentially expressed genes after 9 h (Figure 5c). Of those, 271 differentially expressed genes in the 9 h thermal injury group were also related to *D. melanogaster* genes, as were 198 differentially expressed genes in the 3 and 6 h thermal injury groups. These gene lists were submitted to Metascape for gene enrichment analysis.

### 2.4. Gene Enrichment Analysis of Differentially Expressed Genes

Using Metascape, gene enrichment analysis was performed using the list of differentially expressed genes that were converted to their *D. melanogaster* homologs. Metascape generated 20 genetic functional groups for Gene Ontology (GO) for each thermal injury stimulation group. The GO term (GO:0050830) for defense response to Gram-positive bacteria was enriched at 3, 6, and 9 h after thermal injury (Figure 5d–f). The term defense response to Gram-positive bacteria includes the following AMPs: *defensin* (NCBI ID: NM_078948.3), *attacin-A* (NM_079021.5), *diptericin B* (NM_079063.4), *cecropin A1* (NM_079849.4), *cecropin A2* (NM_079850.4), and *cecropin C* (NM_079852.3). We verified the transcripts per million (TPM) value for these transcripts, which were upregulated by thermal injury stimulation in each group and annotated as AMPs (Appendix A).

Pro-phenoloxidase (PPO) genes (XM_038046567) have been annotated in the BSF genome sequence along with the peptidoglycan-recognition protein-SA (PGRP-SA) gene (XM_038064003). Thus, we checked PPO and PGRP-SA mRNA expression by RNA-seq analysis at 3, 6, and 9 h in the thermal injury groups. In contrast to the non-stimulus control groups, PPO mRNA expression did not fluctuate among the thermal injury groups (Appendix A), but PGRP-SA mRNA expression was found to be increased at 3 h in the thermal injury groups (Appendix A).

Verification of phenoloxidase and peptidoglycan-recognition protein-SA mRNA expression and measurement of phenoloxidase activity in plasma, we examined the mRNA expression of PPO and PGRP-SA through RT-qPCR analysis at 3, 6, 9, and 24 h after thermal injury. PPO mRNA expression did not significantly change among each thermal injury group compared to the non-stimulus control groups (Figure 6a). PGRP-SA mRNA expression was significantly upregulated in the 3 h after thermal injury groups compared to the non-stimulus control groups (Figure 6b).

Next, we measured phenoloxidase activity in the plasma of the thermal injury stimulation group compared to the non-stimulus control group by monitoring the oxidation of the substrate dopamine according to a previously described phenoloxidase assay [28]. There was no significant difference in phenoloxidase activity between the thermal injury and non-stimulus groups (Figure 6c).

## 3. Discussion

As mentioned above, BSF larvae can serve as a dietary protein source whose ability to produce AMPs may decrease the use of antibiotics in animal feed. To construct a safer and easier method to induce AMPs in BSF larvae, we examined several types of stimulation and discovered that thermal injury of larvae activates an innate immune response, resulting in the production of AMPs that are secreted into the hemolymph. The insect’s humoral immune system is stimulated when a pathogen enters the insect’s body. Pathogen recognition proteins recognize the cell wall structure of pathogens [29], which then activate the Toll and IMD pathways [30,31]. Peptidoglycan in the cell wall of bacteria is recognized by peptidoglycan-recognition proteins, which trigger the subsequent immune cascade [15,32]. AMPs and lysozyme are produced within hours in the fat body and hemocytes upon bacterial infection and secreted into the hemolymph to eliminate invading pathogens. As previously mentioned, the BSF humoral immune system can be activated by the injection of living or dead bacteria, or the larvae can be activated by stab wounds in the body [17,19,20,21,22]. However, these methods are not suitable for scaling to the level needed for feeding insect meal to livestock. We examined several types of stimulation expected to induce an immune response, including heat treatment, HCl treatment, thermal injury, stab wounds, and *M. luteus* injection. 

Ambient temperature affects physiological responses and metabolic processes in insects [33]. Takano et al. [34] showed that heat treatment induced an immune response in *Chrysodeixis eriosoma*. Therefore, we expected that heat treatment may induce AMP production. However, heat treatment of the BSF larvae did not significantly induce HiDLP4 and HiCLP1 mRNA expression.

Meanwhile, thermal injury did induce several AMP synthesis. Moreover, this method is both safer and simpler than *M. luteus* injections or stab wounding treatment. 

Next, we performed transcriptome analysis to investigate the AMPs produced by thermal injury, and we found increased attacin, diptericin, defensin, and cecropin mRNA expression at 3, 6, and 9 h after thermal injury, and their mRNA expression pattern differed. Also, we found antimicrobial activity was increased at 9, 15, and 18 h after the thermal injury group. This leads us to speculate that the optimal timing for inducing each AMP protein may differ depending on the specific type of AMP. Consequently, future investigations will be crucial in determining the optimal conditions for maximizing antibacterial activity by determining the precise induction times for each AMP.

Gene enrichment analysis revealed that thermal injury stimulated the expression of a group of genes annotated as part of a defense response to Gram-positive bacteria (GO:0050830). The production of the AMP drosomycin was previously induced by laser wounding in *D. melanogaster* larvae, but this treatment did not induce Diptericin production [35]. Drosomycin is an antifungal peptide [36], and diptericin is a protein active against Gram-negative bacteria [37]. In our study, diptericin, but not drosomycin, transcripts were detected after thermal injury. Thus, it is possible that the thermal damage-stimulating antimicrobial responses in *D. melanogaster* might differ from those in BSF.

Peptidoglycan-recognition protein-SA homolog (PGRP-SA; NM_132499.3, corresponding to MSTRG.8633) was upregulated 3 h after thermal injury, but the protein and mRNA expression of phenol oxidase were unaltered compared to the non-stimulus control groups. Therefore, thermal injury appears to stimulate the expression of a set of AMPs but does not trigger the expression or activation of prophenoloxidase and the melanization response.

Damage-associated molecular patterns (DAMPs) are induced by inflammatory factors mainly derived from cell damage [37]. Several factors contribute to cell damage, including necrosis, the generation of reactive oxygen species (ROS), and tumors [38]. Excessive ROS production produces cell damage via inflammation by DAMPs activation. When DAMPs are released, excessive ROS production can cause inflammation and cell damage [39]. ROS molecules produced during infection are associated with antibacterial responses [38], e.g., the hydrogen peroxide produced by damaged tissues in Drosophila embryos mediates the recruitment of hemocytes [15].

In conclusion, we discovered that thermal injury effectively induces the expression of several types of AMPs in the larvae of *H. illucence* without injecting bacteria. Immunogenic molecules released from cells damaged by thermal injury may stimulate the Toll pathway to elicit the expression of AMPs. However, we were unable to elucidate the specific bactericidal or bacteriostatic effects of each AMP in this study. In our future research, we will examine these actions in AMPs and seek to determine the optimal induction time for each AMP.

Our constructed methods might aid in improving FCR for livestock feed and guard against the emergence of future antibiotic-resistant microorganisms. 

## 4. Materials and Methods

### 4.1. Insects

BSF female adults were obtained from the Fuchu campus at the Tokyo University of Agriculture and Technology, and their oviposited eggs were collected. The eggs were placed on artificial diets with the following composition: 600 g of CLEA Rodent Diet B·F (CLEA Japan, Inc., Tokyo, Japan), 600 g of Rabbit and Guinea Pig Diets LRC4 (ORIENTAL YEAST Co., Ltd., Tokyo, Japan), 30 mL of soybean oil (Fujifilm Wako Pure Chemical Co., Ltd., Osaka, Japan), 6 mL of propionic acid (Fujifilm Wako Pure Chemical Co., Ltd.), 6 g of methyl p-hydroxybenzoate (FUJIFILM Wako Pure Chemical Co., Ltd.), and 2400 mL of distilled water. The hatched larvae were maintained at 27 °C with a 16-h light/8-h dark cycle. The final instar larvae were used in this study.

### 4.2. HCl Treatment of the Larvae

To stress the larval body with acid, BSF larvae (n = 3) were submerged in 0.1 M hydrochloric acid or distilled water (non-stimulus control, n = 3) for 10 min. After 3 h, the fat bodies were dissected from the larvae in each group. Each sample was processed for total RNA purification and then RT-qPCR analysis.

### 4.3. Heat Treatment of the Larvae

To indirectly induce heat stress in the larva, the Dry ThermoUnit (DTU-18 N, TAITEC Co., Ltd., Tokyo, Japan) was preheated to 50 °C. The lid was removed from a 2.0 mL tube (SARSTEDT AG & Co. KG Munich, Germany) using scissors, and the tube was then positioned in the Dry ThermoUnit. BSF larvae (n = 3) were transferred to each 2.0 mL tube and heated for 5 s; non-heated BSF larvae were used as a non-stimulus control (n = 3). After 3 h, the fat bodies were dissected from the larvae in each group. Each sample was processed for total RNA purification and then RT-qPCR analysis.

### 4.4. Stab Wound to the Larvae

To stab wound the larval body, the thorax was pricked once with a 30 G needle (Dentronics Co., Ltd., Tokyo, Japan). Non-pricked larvae were used as a control. The fat bodies of the larvae in each group were dissected after 3 h from the wounded (n = 3) and non-stimulus control (n = 3) groups. Each sample was then processed for total RNA purification and RT-qPCR analysis. The hemolymph of the larvae in each group was collected after 9 (n = 10), 15 (n = 10), 18 (n = 10), 21 (n = 10), or 24 (n = 10) h, and each sample was then processed for the measurement of antimicrobial activity.

### 4.5. Thermal Injury to the Larvae

To induce thermal injury to the larval epidermis, a berry pin (22-735, CLOVER MFG. Co., Ltd., Osaka, Japan) was heated on a gas stove (PA-E18S, Paloma Co., Ltd., Aichi, Japan) for 3 s. The larval thorax was then touched to a heated berry pin for 1 s. Larvae that touched a room-temperature berry pin were used as non-stimulus control. The fat bodies of the larvae in both thermal injury and non-stimulus groups were dissected after 3 (n = 3), 6 (n = 3), 9 (n = 3), or 24 (n = 3) h. Each sample was then processed for total RNA purification, RNA analysis, or RT-qPCR analysis. Each sample was then processed for the measurement of antimicrobial activity or phenoloxidase activity. The hemolymph in both thermal injury and non-stimulus groups was collected after 9 (n = 10), 15 (n = 10), 18 (n = 10), 21 (n = 10), or 24 (n = 10) h for measurement of antimicrobial activity. For measuring phenoloxidase activity, the hemolymph in thermal injury and non-stimulus groups was collected after 9 (n = 4), 15 (n = 5), 18 (n = 5), 21 (n = 5), or 24 (n = 5) h.

### 4.6. Injection of Micrococcus luteus

*M. luteus* was cultured in Luria–Bertani medium (1 g of peptone, 0.5 g of yeast extract, 0.5 g of NaCl, and 0.1 g of glucose per 100 mL of distilled water) at 30 °C for 16 h. Cells in the logarithmic growth phase were harvested by centrifugation at 1800× *g* for 20 min at 4 °C, washed twice with insect physiological saline (IPS; 150 mM NaCl, 5 mM KCl, and 1 mM CaCl_2_), and fixed with 4% formaldehyde by gentle shaking for 1 h. The fixed cells were subsequently harvested by centrifugation at 1800× *g* for 20 min at 4 °C and washed 5 times with Clark’s saline (110 mM NaCl, 188 mM KCl, 1 mM CaCl_2_, 1 mM NaHCO_3_, and 0.07 mM Na_2_HPO_4_). Insect physiological saline was used to suspend fixed *M. luteus* to a McFarland standard turbidity of 0.5. The final instar BSF larvae were anesthetized on ice, and 10 μL of fixed *M. luteus* was injected into them using a 1 mL sterile syringe (TERUMO Co., Ltd., Tokyo, Japan) with a 30 G needle (Dentronics Co., Ltd., Tokyo, Japan). We induced an immune response by injecting *M. luteus* or 10 μL of PBS containing 4 mM glutathione (in insect phosphate buffered saline; non-stimulus group). After 3 h, the larvae were swabbed with 70% ethanol, and then fat bodies were dissected from the larvae, and then the sample was processed for total RNA purification and RT-qPCR. The hemolymph of the larvae was collected after 9 (n = 10), 15 (n = 10), 18 (n = 10), 21 (n = 10), or 24 (n = 10) h. Each sample was then processed for the measurement of antimicrobial activity.

### 4.7. Total RNA Purification and cDNA Synthesis from Fat Body Samples and RT-qPCR Analysis

The fat bodies from each group were used for total RNA purification. These samples were stored at −80 °C until use. The fat bodies were homogenized in 1 mL of TRIzol^TM^ reagent (Thermo Fisher Scientific, Waltham, MA, USA). Then, 200 µL chloroform was added, and tubes were vortexed for 15 s. After 3 min, the aqueous phase was collected after centrifugation at 12,000× *g* for 10 min at 4 °C. The supernatant was transferred to a new 1.5 mL tube, then 500 µL of isopropanol was added, and tubes were vortexed for 15 s. After 10 min, precipitated total RNA was collected after centrifugation at 12,000× *g* for 10 min. The pellet was collected and resuspended in 1 mL of 75% ethanol. Then, the supernatant was removed after centrifugation at 12,000× *g* for 10 min. This step was repeated twice. The collected RNA pellet was air-dried for 5–10 min. Finally, the RNA pellet was resuspended in 50 µL of RNase-free water, and RNA concentration was measured using NanoDrop One^TM^ (Thermo Fisher Scientific Inc., Valencia, CA, USA). 

One microgram of total RNA was treated with DNase I (Invitrogen, Van Allen Way, Carlsbad, CA, USA), and then 500 ng of DNase-treated total RNA was used as a template for cDNA synthesis using a PrimeScript™ 1st strand cDNA Synthesis Kit (Takara Co., Ltd., Tokyo, Japan) in accordance with the manufacturer’s instructions. Real-time quantitative PCR (RT-qPCR) was performed in 20 μL reaction volumes with 0.5 μL of cDNA template and the specific primers (Appendix A) along with a KAPA SYBR Fast qRT-PCR Kit (Nippon Genetics Co., Ltd., Tokyo, Japan) in accordance with the manufacturer’s instructions.

RT-qPCR was performed on Step One Plus Real-Time PCR System (Applied Biosystems, Foster City, CA, USA) by following the Delta–Delta Ct method. The BSF ribosomal RNA 18 gene (*rs18*, Gene ID LOC11965445) was utilized as an endogenous reference against which RNA expression levels were standardized, and all data were calibrated against universal reference data. The relative expression level in comparison to a reference sample is represented by relative quantification (RQ) values. RQ values represent the relative expression levels calculated using each non-stimulus sample as 1. Error bars represent the relative minimum/maximum expression levels of the mean RQ value. All sample sets were assayed in triplicate as biological replicates.

### 4.8. cDNA Cloning

The PCR amplified cDNA products with the specific primers (Appendix A) were cloned using a TOPO^®^ TA Cloning^®^ Kit (Thermo Fisher Scientific, Waltham, MA, USA) and used to transform ECOS™ Competent *E. coli* XL-1 blue (NIPPON GENE Co., Ltd., Tokyo, Japan) for subcloning. Then, using a DNA analyzer by Applied Biosystems, the 3730xl nucleotide sequences of the cDNA inserts were determined (Thermo Fisher Scientific).

### 4.9. Hemolymph Collection for Phenoloxidase Activity

BSF larvae were anesthetized on ice. The larvae were swabbed with 70% ethanol and nicked between the eighth and ninth abdominal segments using microscissors. The hemolymph was immediately collected in a 1.5 mL tube on ice using a micropipette (P10L, Gilson Co., Ltd. Middleton, WI, USA). Then, the collected hemolymph was centrifuged for 10 min at 12,000× *g* at 4 °C to obtain the plasma. The plasma samples were stored at −30 °C until use.

### 4.10. Measurement of Phenoloxidase Activity

Larval hemolymph plasma was used to measure phenoloxidase activity. Two microlitres of plasma were added to a 1.5 mL tube containing either 1 μL of filter-sterilized deionized water (control) or 1 μL of M. luteus suspension (1 μg/μL suspension, Sigma–Aldrich), and the tubes were incubated for 10 min at room-temperature (RT). Following incubation, these plasma samples were divided among 96-well plates. Each well then received 200 μL of 2-mM dopamine hydrochloride (Sigma–Aldrich) solution dissolved in a 50-mM sodium phosphate buffer with a pH of 6.5. The absorbance at 470 nm was monitored once every minute for 30 min at 30 °C using a microplate spectrophotometer (Synergy H1, Agilent BioTek, Santa Clara, CA, USA). The reaction curves started with a lag phase followed by a linear phase. The changes in absorbance per minute during the linear phase of each reaction were calculated using linear regression. One unit of phenoloxidase activity was defined as ΔA470 = 0.001/min. The assay was performed in duplicate.

### 4.11. RNA-Seq Analysis

The fat bodies were collected at 3, 6, and 9 h after thermal injury (n = 3 in each group), along with their corresponding non-stimulus control samples (n = 3 in each group). Then, these fat body samples were used for total RNA purification. The RNA samples were used for the RNA-seq analysis of samples taken 3, 6, and 9 h after thermal injury (n = 3 in each group), along with their corresponding non-stimulus control samples (n = 3 in each group). RNA quality was assessed using TapeStation 2200 software (Agilent Technologies, Inc., Santa Clara, CA, USA). Additionally, cDNA libraries for paired-end sequencing were constructed with 100 ng of total RNA using a NovaSeq^®^ 6000 SP Reagent Kit (Illumina, Inc., San Diego, CA, USA) from each control and thermal injury group (n = 3 each) according to the manufacturer’s instructions. The libraries were sequenced (101 bp, paired-end) on the Illumina NovaSeq6000 platform, and FASTQ files were assessed with the Trim Galore! (v0.6.7) trimming tool (https://www.bioinformatics.babraham.ac.uk/projects/trim_galore/ accessed on 11 May 2020). The BSF genome (iHerIll2.2.curated.20191125) sequence was retrieved from the NCBI Genome database (https://www.ncbi.nlm.nih.gov/assembly/GCF_905115235.1 accessed on 26 January 2022). The obtained FASTQ sequence files were aligned to the genomic reference sequence using the HISAT2 v2.2.1 alignment program for mapping RNA-seq reads with the default parameters [40]. Next, the obtained SAM files were converted to BAM files with SAMtools v1.14 [41]. Transcript abundance was estimated using the StringTie v2.1.7 assembler, and the count data were extracted with the Subread v1.6.0 read aligner [42]. All statistical analyses were performed using R software version v3.6.3 (https://www.r-project.org accessed on 20 January 2022). The data were normalized, and the control and thermal injury groups were compared using the TCC and DEseq2 packages [43]. R software was used to create an MA plot. The RNA-seq datasets are available in the Sequence Read Archive under accession numbers DRR426824-DRR426841. Appendix A presents the sample information.

### 4.12. Gene Enrichment and Molecular Interaction Analyses

Gene enrichment analyses were performed using the Metascape gene annotation and analysis resource39 (http://metascape.org/ accessed on 31 January 2022). A gene list for Metascape analysis was generated from the TCC output. The gene ID numbers were converted from the *H. illucence* RNA-seq data to *D. melanogaster* NCBI ID numbers with an e-value (1E-10) for the construction of an assignment table. Then, the list of genes obtained from the RNA-seq data was entered into the IntAct Molecular Interaction Database [44] to identify significant molecular interactions.

### 4.13. Hemolymph Collection for Antimicrobial Activity Assay

BSF larvae (n = 10) were anasthetized on ice. Then 0.1 mL of 4 mM glutathione in insect phosphate-buffered saline (to block melanization) was injected into them using a 1 mL sterile syringe (TERUMO Co., Ltd.) with a 30 G needle (Dentronics Co., Ltd.). 

The larvae were swabbed with 70% ethanol and nicked between the eighth and ninth abdominal segments using microscissors. Hemolymph drops were collected in a 1.5 mL tube directly and centrifuged for 10 min at 12,000× *g* at 4 °C to obtain plasma. To avoid the influence of immune factors other than AMPs, we heated the plasma for 30 min at 60 °C, followed by centrifugation for 10 min at 12,000× *g* at 4 °C. The supernatant (plasma) was immediately used in assays of antimicrobial activity. 

### 4.14. Antimicrobial Activity Assay

An *M. luteus* (ATCC 4698TM) glycerol stock was precultured in tryptic soy broth (Becton Dickinson GmbH, Heidelberg, Germany) for the assay. The optical densities at 600 nm were adjusted to 0.26 by adding a tryptic soy broth culture medium. Hemolymph samples were filtered with a 0.2-μm sterilized filter (Merck Millipore Co., Ltd., Burlington, MS, USA), and the total protein concentration was then diluted to 1 mg/mL using 4 mM glutathione (Fujifilm Wako Pure Chemical Co., Ltd.) in insect PBS solution at pH 6.5. One hundred microliters of the culture medium and 10 μL of the hemolymph samples were transferred to a 96-well round bottom microwell plate (#351177, Falcon^®^, Corning Co., Ltd., Corning, NY, USA). A total of 10 μL of 4 mM glutathione in insect PBS solution at pH 6.5 was used as a negative control. A total of 10 μL of 0.1 mg/mL ampicillin was used as a positive control. The absorbance at 600 nm was monitored once per minute for 42 h at 37 °C in shaking mode using a microplate spectrophotometer (Synergy H1, Agilent BioTek). Lag, log, stationary, and death phases constitute four bacterial growth phases that make up the bacterial growth curve. If the hemolymph contained sufficient AMPs, the lag phase growth was extended. Thus, we concentrated on the period when growth reached the log phase as a measure of antibacterial activity. The time for the stay of the lag phase (h) was calculated using the tangent hyperbolic function by Microsoft Excel, and then we expressed extended lag phase growth as the time of stay in the lag phase (hours). The assay was performed in triplicate.

### 4.15. Statistical Analysis

Statistical significance was assessed using a two-tailed Student’s *t*-test in Excel (Microsoft, Redmond, WA, USA). *p* values < 0.05 were considered significant.

## Figures and Tables

**Figure 1 ijms-24-15765-f001:**
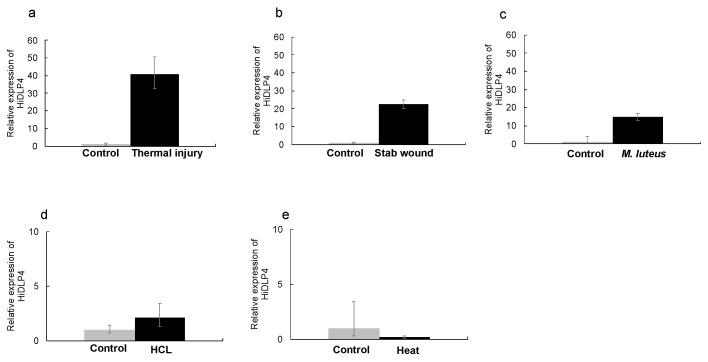
Expression of HiDLP4 mRNA in response to several types of stimulation of the last instar larvae. Several types of stimulation were given to the last instar larvae. After 3 h, the fat bodies were extracted from the larvae in each group, and then each sample was processed for RT-qPCR analysis. (**a**) Thermal injury (touched with a heated pin for 1 s), (**b**) stab wound, (**c**) *M. luteus* injection, (**d**) HCl stimulation (larvae submerged in 0.1 M hydrochloric acid for 10 min), and (**e**) heat treatment (incubation at 50 °C for 5 s). These samples’ RQ values represent the levels of mRNA expression in the fat body. RQ values were calculated relative to expression levels for each non-stimulus sample, normalized as 1. Error bars represent the relative minimum/maximum expression levels of the mean RQ value (3 biological replicates). *Hermetia illucens* ribosomal protein s18 (*Hirs18*) was used as the endogenous control.

**Figure 2 ijms-24-15765-f002:**
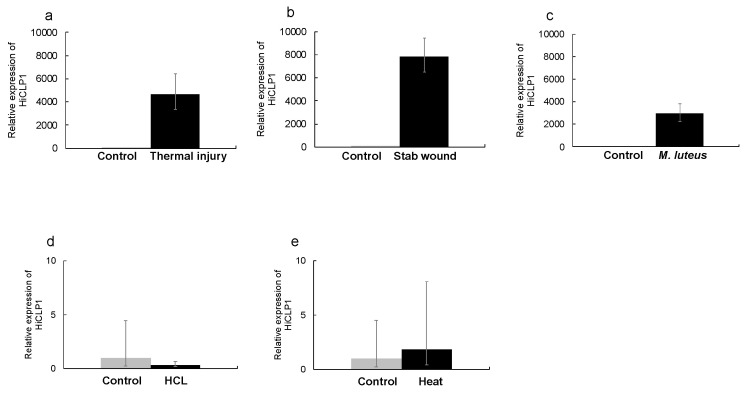
Expression of HiCLP1 mRNA in response to different types of stimulation. Several types of stimulation were given to the last instar larvae. After 3 h, the fat bodies were extracted from the larvae in each group, and then each sample was processed for RT-qPCR analysis. (**a**) Thermal injury, (**b**) stab wound, (**c**) *M. luteus* injection, (**d**) HCl stimulation, and (**e**) heat treatment. These samples’ RQ values represent the levels of mRNA expression in the fat body. RQ values were calculated relative to expression levels for each non-stimulus sample normalized as 1. Error bars represent the relative minimum/maximum expression levels of the mean RQ value (3 biological replicates). *Hirs18* was used as the endogenous control.

**Figure 3 ijms-24-15765-f003:**
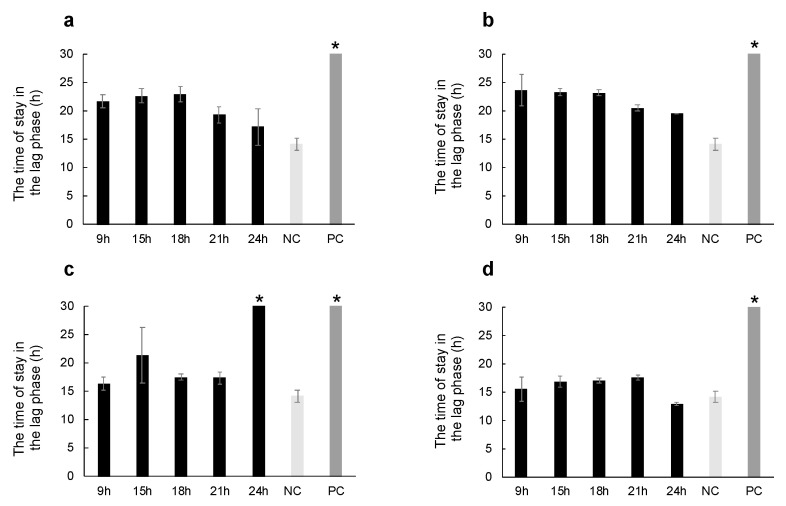
Comparison of antibacterial activity in larvae subjected to thermal injury and other treatments. BSF larvae were subjected to thermal injury, stab wounds, or *M. luteus* injection. Larvae were touched with a room-temperature berry pin as non-stimulus control. Antibacterial activity in the larval plasma was compared among the (**a**) thermal injury, (**b**) stab wound, (**c**) *M. luteus* injection, and (**d**) non-stimulus control groups. The graph shows the time of the lag phase (hours). An asterisk indicates that the assays failed to reach the log phase within 42 h due to high antibacterial activity. PC indicates positive control, 10 μL of 1 mg/mL ampicillin. NC indicates negative control, 10 μL of PBS containing 4 mM glutathione. The antibacterial activity represents the mean ± SD. The assay was performed in triplicate.

**Figure 4 ijms-24-15765-f004:**
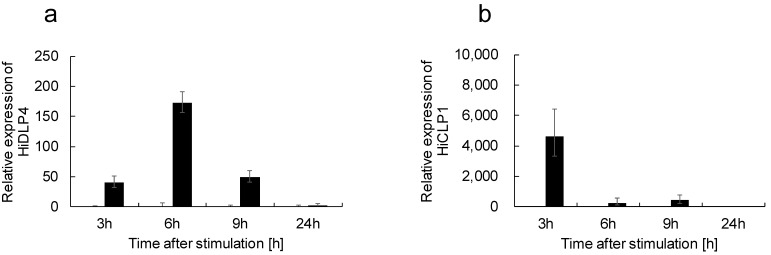
Time course of HiDLP4 and HiCLP1 mRNA expression in response to thermal injury of last instar larvae. At 3, 6, 9, and 24 h after thermal injury of the last instar larvae, the fat bodies were extracted from the larvae (n = 3) in each group, and then each sample was processed for RT-qPCR analysis. (**a**) HiDLP4, (**b**) HiCLP1. The mRNA expression in the fat body of these samples is the RQ value, representing the relative expression levels calculated using non-stimulus larvae normalized as 1. Error bars represent the relative minimum/maximum expression levels of the mean RQ value (3 biological replicates). *Hirs18* was used as the endogenous control.

**Figure 5 ijms-24-15765-f005:**
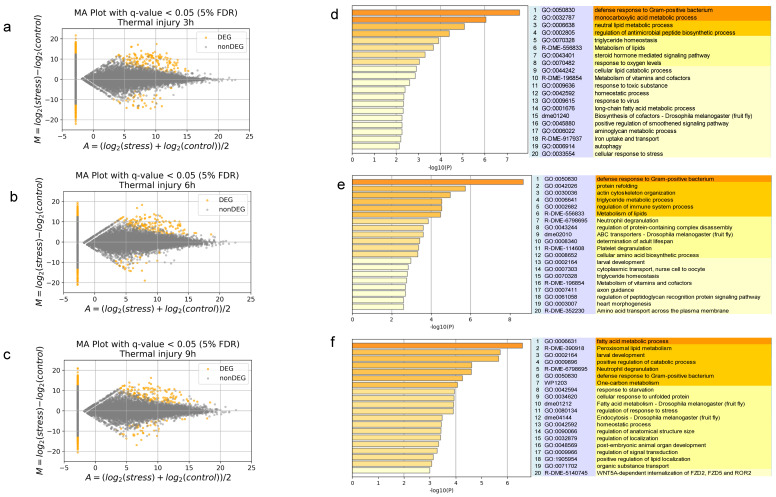
MA plots and gene enrichment analyses of samples at 3 h, 6 h, and 9 h after thermal injury. M represents the log ratio, and A represents the mean average scale. Orange-colored dots indicate differentially expressed transcripts with a false discovery rate [FDR] < 0.05. (**a**) 3 h, (**b**) 6 h, and (**c**) 9 h after thermal injury. Gene enrichment analysis of differentially expressed transcripts in thermal injury stimulation using Metascape. A bar graph of enriched terms across the input transcript lists; different colored bars indicate *p* values. (**d**) 3 h, (**e**) 6 h, and (**f**) 9 h after thermal injury.

**Figure 6 ijms-24-15765-f006:**
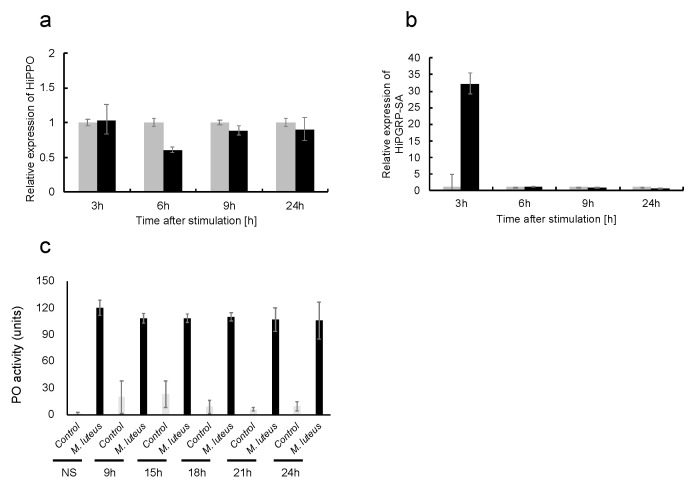
mRNA expression analyses of pro-phenoloxidase and peptidoglycan-recognition protein-SA and phenoloxidase activity assay of plasma samples from BSF larvae after thermal injury. Thermal injury stimulation was performed on the last instar larvae. After 3, 6, 9, and 24 h, the fat bodies were extracted from the larvae (n = 3) in each group, and then each sample was processed for RT-qPCR analysis. (**a**) PPO mRNA expression. The mRNA expression in the fat body of these samples is RQ values. RQ values represent the relative expression levels calculated with non-stimulus larvae normalized as 1. Error bars represent the relative minimum/maximum expression levels of the mean RQ value (3 biological replicates). Hirs18 was used as the endogenous control. (**a**) PPO mRNA expression and (**b**) PGRP-SA mRNA expression. At 9, 15, 18, 21, and 24 h after thermal injury of the last instar larvae, plasma was collected from the larvae in each group, and then each sample was processed for a phenoloxidase (PO) assay. Control, no added bacteria; *M. luteus* indicates plasma samples treated with *M. luteus* prior to the PO assay. Error bars indicate standard deviation (n = 10). The assay was performed in duplicate. (**c**) PO activity. Statistical significance was calculated by comparing it to the non-stimulus control. 9 h (*p* = 0.046), 15 h (*p* = 0.024), 18 h (*p* = 0.040), 21 h (*p* = 0.086), 24 h (*p* = 0.186).

## Data Availability

The RNA-seq datasets are available in the Sequence Read Archive under accession numbers DRR426824-DRR426841. Appendix A presents the sample information. GeneBank/DDBJ SAKURA Data bank under accession numbers LC741404 (for HiDLP4) and LC741405 (for HiCLP1).

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
