# Peer review of "The Development of New Methods to Stimulate the Production of Antimicrobial Peptides in the Larvae of the Black Soldier Fly Hermetia illucens"

_ijms, 2023, doi:10.3390/ijms242115765_

Round 1

Reviewer 1 Report

Comments and Suggestions for Authors

The MS by Nakagawa et al. tests various possible ways to activate an expression of antimicrobial peptides (AMP) in Hermetia illucens larvae: thermal injury (touched with a heated pin for 1 s), stab wound and acute heat stress in comparison with traditional way of AMP activation, bacteria injection. The authors found out that the thermal injury only is able to induce the expression of several types of AMPs effectively.

The study is well setup, looks well executed and contains new data which are well discussed. The manuscript could be of interest to the readership of IJMS.

Minor comments.

1. It takes some time to understand what “Hirs18” in RT-PCR experiments mean. As the Materials & Methods section is in the end of the paper, it would be better to explain the gene name in the Results.

2. Lines 503-505. There is an extra verb in this sentence:

“To avoid the influence of immune factors other than AMPs, we treated the plasma was heated for 30 min at 60 °C and then centrifuged it for 10 min at 12,000 g at 4 °C.”

Comments on the Quality of English Language

I believe that the Abstract needs some editing in order to improve its readability.

Author Response

Dear Reviewer #1

Comment: The MS by Nakagawa et al. tests various possible ways to activate an expression of antimicrobial peptides (AMP) in Hermetia illucens larvae: thermal injury (touched with a heated pin for 1 s), stab wound and acute heat stress in comparison with traditional way of AMP activation, bacteria injection. The authors found out that the thermal injury only is able to induce the expression of several types of AMPs effectively. The study is well setup, looks well executed and contains new data which are well discussed. The manuscript could be of interest to the readership of IJMS.

Answer: Thank you for positive comments. We appreciate your thoughtful review of our manuscript. Your feedback has been invaluable in improving the quality of our work. We have carefully considered your suggestion (below) and made the necessary revisions. We believe that these changes have strengthened the overall quality and clarity of our manuscript.

Minor comments.

Suggestion 1: It takes some time to understand what “Hirs18” in RT-PCR experiments mean. As the Materials & Methods section is in the end of the paper, it would be better to explain the gene name in the Results.

Answer 1: Thank you for this suggestion. In the Results section, we added the full gene name for Hirs18: Hermetia illucens ribosomal protein s18 (line 133).

Suggestion 2:. Lines 503-505. There is an extra verb in this sentence:

“To avoid the influence of immune factors other than AMPs, we treated the plasma was heated for 30 min at 60 °C and then centrifuged it for 10 min at 12,000 g at 4 °C.”

Answer 2: We removed extra verb. The revised sentence is as follows: “To avoid the influence of immune factors other than AMPs, we heated the plasma for 30 min at 60°C, followed by centrifugation for 10 min at 12,000 g and 4°C.” (lines 457–459).

Suggestion 3: I believe that the Abstract needs some editing in order to improve its readability.

Answer 3: Thank you for your comment. We have revised the abstract to enhance its readability as follows:

“The global population is projected to reach a staggering 9.8 billion people by the year 2050, leading to major concerns about food security. The necessity to increase livestock production is inevitable. The black soldier fly (BSF) is known for its ability to consume a wide range of organic waste, and BSF larvae have already been used as a partial substitute for fishmeal. In contrast, the use of antibiotics in livestock feed for growth promotion and prophylaxis poses a severe threat to global health owing to antimicrobial resistance. Insect antimicrobial peptides (AMPs) have shown the potential to rapidly disrupt target bacterial membranes, making bacterial resistance to AMPs a less likely concern. (2) Methods: In this study, we explored various methods for stimulating AMP synthesis in BSF larvae and found that thermal injury effectively induced the production of various AMP types. Additionally, we investigated the activation of innate immune response pathways that lead to AMP production following thermal injury. (3) Results: Interestingly, thermal injury treatment, although not involving bacteria, exhibited a similar response to that observed following gram-positive bacterial infection in eliciting the expression of AMP genes. (4) Conclusions: Our findings offer support for the industrial use of BSF to enhance livestock production and promote environmental health.” (lines 17-31).

Reviewer 2 Report

Comments and Suggestions for Authors

I read the manuscript titled "The development of new methods to stimulate the production of antimicrobial peptides in the larvae of the black soldier fly, Hermetic illucens" with great interest. The authors describe of developing alternative methods for stimulating the production of AMPs in black soldier flies without the use of bacteria. They conclude that thermal injury elicited the expression of AMP genes and was in par with bacterial infection. Although the manuscript deals with an important topic, there are some major flaws in the methods used by the authors. I have provided some suggestions to the authors below so that they could improve upon the manuscript.

Line 33-50: Introduction: The 1st and 2nd para do not introduce the topic well to the general readers. I advice the authors to re-write the 1st and 2nd paragraphs of the introduction so that the general readers are better introduced about the topic.

Line 94: Did the authors really examine the production of AMPs? or was it just the expression of mRNAs? The authors should note that the expression of mRNAs is not the same as production of proteins. The mRNA may get expressed in large quantities, but there are many other factors that may hinder the translation of mRNA into proteins. Both are not the same. Please see Greenbaum, D., Colangelo, C., Williams, K. et al. Comparing protein abundance and mRNA expression levels on a genomic scale. Genome Biol 4, 117 (2003). https://doi.org/10.1186/gb-2003-4-9-117. While mRNA expression values have shown their usefulness in a broad range of applications, including the diagnosis, these results are almost certainly only correlative. In the end it is most probably the concentration of proteins and their interactions that are the true causative forces in the cell, and it is the corresponding protein quantities that we ought to be studied. I did not see the authors isolate the AMPs or did I overlook this? As this study is about AMPs that have a causative effect of anti-microbial activity, it would be best to check for the conc. of AMPs rather than just the mRNAs.

Line 117-118: Why was the heat treatment for one a singe temperature, 50 deg C? Testing at different various temperature may have given a better perspective to the study.

Line 146-147: What is the basis of using the bacteria M. luteus? Is it a pathogen of the soldier fly?

Line 154: What were the authors testing, bactericidal or bacteriostatic activity? Extended lag phase studies can show bacteriostatic effects but not bactericidal. So if the authors were checking for bacteriostatic effects of AMPs and if this is true, then the use of ampicillin is not justified as ampicillin is bactericidal antibiotic. The authors could have used tetracyclines as they are bacteriostats.

As the work is much needed, I advice the author to look into the comments carefully and conduct the experiments as suggested to improve the impact of the manuscript.

Author Response

Dear Reviewer #2

Comment: I read the manuscript titled "The development of new methods to stimulate the production of antimicrobial peptides in the larvae of the black soldier fly, Hermetic illucens" with great interest. The authors describe of developing alternative methods for stimulating the production of AMPs in black soldier flies without the use of bacteria. They conclude that thermal injury elicited the expression of AMP genes and was in par with bacterial infection. Although the manuscript deals with an important topic, there are some major flaws in the methods used by the authors. I have provided some suggestions to the authors below so that they could improve upon the manuscript.

Answer: Thank you for your valuable comments and suggestions. We have thoroughly reviewed and incorporated your constructive feedback into our manuscript. We are confident that these revisions have markedly enhanced the quality of our work and addressed the points you raised in your review.

Suggestion 1: Line 33-50: Introduction: The 1st and 2nd para do not introduce the topic well to the general readers. I advice the authors to re-write the 1st and 2nd paragraphs of the introduction so that the general readers are better introduced about the topic.

Answer 1: Thank you for your comments and suggestion. We have made several improvements to the manuscript to ensure it is more accessible to a general readership. Specifically, we used the common name of the insect rather than the scientific name, and we have revised the first and second paragraph to improve their general clarity and readability as follows:

“The black soldier fly (BSF; Hermetia illucens) is a dipteran insect known for its distinctive characteristics. BSF larvae can consume a wide range of organic waste materials, including food scraps from various sources and animal excrement [1]. Owing to these unique attributes, the BSF has garnered attention as a promising organism for the sustainable production of protein resources from organic waste [2]. With the global population projected by the United Nations to reach 9.77 billion by 2050 [3], the challenge of ensuring adequate food supply for this growing population is becoming increasingly important. To address this issue, there is a growing need to enhance livestock production, which, in turn, raises concerns about potential shortages in livestock feed.

In Japan, various livestock, such as cows, pigs, and chickens, are traditionally reared on a diet composed of mixture of concentrate (comprising soybean, corn, wheat, fishmeal, skim milk, and other components) and roughage (including fresh grass, silage, hay, and straw) [4]. In recent years, BSF larvae have been used as a partial substitute for fishmeal in livestock feed [5–7]. Furthermore, the routine use of antibiotics in livestock farming, aimed at promoting growth, has become a topic of increasing concern [8–11]. Excessive antibiotic use poses serious societal risks, as it contributes to the emergence of antibiotic-resistant bacteria and their subsequent spread in the environment, jeopardizing public health [12]. Consequently, efforts are underway to identify alternative solutions that can replicate the growth-promoting effects of antibiotics while mitigating these associated risks.” (lines 35–54).

Suggestion 2:Line 94: Did the authors really examine the production of AMPs? or was it just the expression of mRNAs? The authors should note that the expression of mRNAs is not the same as production of proteins. The mRNA may get expressed in large quantities, but there are many other factors that may hinder the translation of mRNA into proteins. Both are not the same. Please see Greenbaum, D., Colangelo, C., Williams, K. et al. Comparing protein abundance and mRNA expression levels on a genomic scale. Genome Biol 4, 117 (2003). https://doi.org/10.1186/gb-2003-4-9-117. While mRNA expression values have shown their usefulness in a broad range of applications, including the diagnosis, these results are almost certainly only correlative. In the end it is most probably the concentration of proteins and their interactions that are the true causative forces in the cell, and it is the corresponding protein quantities that we ought to be studied. I did not see the authors isolate the AMPs or did I overlook this? As this study is about AMPs that have a causative effect of anti-microbial activity, it would be best to check for the conc. of AMPs rather than just the mRNAs.

Answer 2: Thank you for bringing this to our attention. As you are aware, it is common for mRNA expression and protein expression to exhibit variations in correlation. In our study, we observed a significant increase in AMP mRNA expression following thermal injury treatment. Additionally, we noted a rise in antimicrobial activity in the thermal injury treatment group compared with the nonstimulus control. These findings suggest a plausible connection between the induction of AMP protein and the corresponding level of AMP mRNA expression. Our transcriptome data revealed the induction of various types of AMPs upon thermal injury, and they also indicated the induction of AMP mRNA expression at various times after thermal injury, depending on their specific type (Figs. S1–S5). This leads us to speculate that the optimal timing for inducing each AMP protein may differ depending on the specific type of AMP. Consequently, future investigations will be crucial in determining the optimal conditions for maximizing antibacterial activity by determining the precise induction times for each AMP (lines 261–264).

Suggestion 3: Line 117-118: Why was the heat treatment for one a singe temperature, 50 deg C? Testing at different various temperature may have given a better perspective to the study.

Answer 3: Thank you for pointing out this issue. Notably, being poikilothermic (cold-blooded animals), insects lack the ability to regulate a constant internal body temperature, unlike mammals. To address this, we conducted a preliminary experiment to assess the impact of different heat treatments (40°C, 50°C, and 60°C). Our findings revealed that exposure to 60°C markedly weakened the BSF larvae. Consequently, we opted for a treatment temperature of 50°C in this study to ensure the well-being of the study animals.

Suggestion 4: Line 146-147: What is the basis of using the bacteria M. luteus? Is it a pathogen of the soldier fly?

Answer 4: Park et al. previously reported that HiDLP4 and HiCLP1 exhibit antimicrobial activity against the gram-positive bacterium methicillin-resistant S. aureus1). Notably, M. luteus is also a gram-positive bacterium. Additionally, we confirmed a significant increase in HiDLP4 and HiCLP1 mRNA expression levels following thermal injury stimulation. As a result, we selected M. luteus as our chosen bacterium for assessing antimicrobial activity. This information is appropriately referenced in our manuscript as follows:

“Park et al. reported that HiDLP4 exerts antimicrobial activity against the gram-positive bacterium methicillin-resistant S. aureus [20]. Hence, we employed a gram-positive bacterium, M. luteus, to characterize antimicrobial activity.” (lines 128–130).

  • Park S. I., Kim J. W., Yoe S. M., Purification and characterization of a novel antibacterial peptide from black soldier fly (Hermetia illucens) larvae. Dev. Comp. Immunol. 2015, 52, 98-106.

Suggestion 5: Line 154: What were the authors testing, bactericidal or bacteriostatic activity? Extended lag phase studies can show bacteriostatic effects but not bactericidal. So if the authors were checking for bacteriostatic effects of AMPs and if this is true, then the use of ampicillin is not justified as ampicillin is bactericidal antibiotic. The authors could have used tetracyclines as they are bacteriostats.

Answer 5: Thank you for your feedback. Typically, antibacterial activity is assessed using the paper disc diffusion assay, among other possible methods, where the effectiveness of antibacterial substances in inhibiting bacterial growth is measured based on the size of the inhibition zone. As you may be aware, the bacterial growth curve comprises four distinct phases: lag, log, stationary, and death phases. In the presence of sufficient AMPs in BSF hemolymph, the lag phase is extended due to the inhibition of bacterial growth. For this reason, we focused on the period when bacterial growth reached the log phase as an indicator of antibacterial activity. Our antibacterial activity assay, similar to the paper disc method, does not distinguish between bactericidal or bacteriostatic effects. We acknowledge the need to explore the bactericidal or bacteriostatic actions of the AMPs induced by thermal injury treatment in future studies.

We have also incorporated the following sentence into the conclusion section to address this concern:

“However, we were unable to elucidate the specific bactericidal or bacteriostatic effects of each AMP in this study. In our future research, we will examine these actions in AMPs, and seek to determine the optimal induction time for each AMP.” (lines 290–293).

Suggestion 6: As the work is much needed, I advice the author to look into the comments carefully and conduct the experiments as suggested to improve the impact of the manuscript.

Answer 6: Thank you for all of your valuable suggestions. We have incorporated explanations based on our preliminary data findings, further enhancing the quality of our manuscript. We are confident that these additions have strengthened our work and improved its overall quality.

Round 2

Reviewer 2 Report

Comments and Suggestions for Authors

The authors have made substantial improvements to the manuscript and have provided a logical rebuttal to my queries. The manuscript can be published now.